# A Genetically Engineered Commercial Chicken Line Is Resistant to Highly Pathogenic Avian Leukosis Virus Subgroup J

**DOI:** 10.3390/microorganisms9051066

**Published:** 2021-05-14

**Authors:** Ahmed Kheimar, Romina Klinger, Luca D. Bertzbach, Hicham Sid, You Yu, Andelé M. Conradie, Benjamin Schade, Brigitte Böhm, Rudolf Preisinger, Venugopal Nair, Benedikt B. Kaufer, Benjamin Schusser

**Affiliations:** 1Institute of Virology, Freie Universität Berlin, 14163 Berlin, Germany; ahmed.kheimar@fu-berlin.de (A.K.); luca.bertzbach@fu-berlin.de (L.D.B.); yuyou@zedat.fu-berlin.de (Y.Y.); andele.conradie@fu-berlin.de (A.M.C.); 2Department of Poultry Diseases, Faculty of Veterinary Medicine, Sohag University, 82424 Sohag, Egypt; 3Reproductive Biotechnology, Department of Molecular Life Sciences, TUM School of Life Sciences, Technical University Munich, 85354 Freising, Germany; romina.klinger@tum.de (R.K.); hicham.sid@tum.de (H.S.); 4Bavarian Animal Health Service, Department of Pathology, 85586 Poing, Germany; benjamin.schade@tgd-bayern.de (B.S.); brigitte.boehm@tgd-bayern.de (B.B.); 5EW Group GmbH, 49429 Visbek, Germany; rudolf.preisinger@ew-group.de; 6The Pirbright Institute, Woking GU24 0NF, UK; venugopal.nair@pirbright.ac.uk

**Keywords:** avian retrovirus, avian leukosis virus subgroup J, HPRS-103, CRISPR/Cas9, gene editing, chNHE1, resistance, viral escape

## Abstract

Viral diseases remain a major concern for animal health and global food production in modern agriculture. In chickens, avian leukosis virus subgroup J (ALV-J) represents an important pathogen that causes severe economic loss. Until now, no vaccine or antiviral drugs are available against ALV-J and strategies to combat this pathogen in commercial flocks are desperately needed. CRISPR/Cas9 targeted genome editing recently facilitated the generation of genetically modified chickens with a mutation of the chicken ALV-J receptor Na^+^/H^+^ exchanger type 1 (chNHE1). In this study, we provide evidence that this mutation protects a commercial chicken line (NHE1ΔW38) against the virulent ALV-J prototype strain HPRS-103. We demonstrate that replication of HPRS-103 is severely impaired in NHE1ΔW38 birds and that ALV-J-specific antigen is not detected in cloacal swabs at later time points. Consistently, infected NHE1ΔW38 chickens gained more weight compared to their non-transgenic counterparts (NHE1W38). Histopathology revealed that NHE1W38 chickens developed ALV-J typical pathology in various organs, while no pathological lesions were detected in NHE1ΔW38 chickens. Taken together, our data revealed that this mutation can render a commercial chicken line resistant to highly pathogenic ALV-J infection, which could aid in fighting this pathogen and improve animal health in the field.

## 1. Introduction

Avian leukosis viruses (ALVs) are common retroviruses in domestic chickens associated with neoplasia and immunosuppression [1]. ALVs comprise ten distinct subgroups including ALV subgroup J (ALV-J) which currently represents the most prevalent in the field [2]. ALV-J infections often progress subclinical and lead to a severe reduction in egg and meat production [3,4]. Myeloid leukosis (ML), the primary neoplastic disorder induced by subgroup J, is characterized by various tumor types with a high incidence of renal tumors and gross skeletal myelocytomas [1,5].

Especially during the late 1990s, the virus has spread rapidly through the transportation of poultry and poultry products and increased horizontal transmission rates [6]. A reduction in ALV-J outbreaks was only achieved by strict eradication measures [6]. However, ALV-J continues to be a major issue for the poultry industry as eradication programs did not eliminate the virus across the globe. In addition, other common antiviral measures including vaccination or trait selective-breeding were not effective enough [7,8]. ALV-J still causes enormous economic losses in Asia where broiler and laying hen husbandries are severely affected [2]. This became especially apparent during the last major ALV-J outbreak in China in 2018 [9]. Other susceptible species of birds could act as a potential reservoir for ALV-J strains and pose a risk of pathogen transmission [10,11].

Recent advances in genome editing technology have expanded the toolbox of antiviral control strategies [12]. Here, CRISPR/Cas9-based genome editing serves as a precise and efficient molecular tool to modify host genes that play a key role in the viral replication cycles and render the host resistant against the pathogen [12,13]. We and others recently used this strategy to generate ALV-J resistant chickens [14,15]. Virus entry was inhibited by a precise modification of the ALV-J host cell receptor, chicken Na^+^/H^+^ exchanger type 1 (chNHE1), encoded by the *tvj* locus on chromosome 23 [16]. The ALV-J binding site consists of 12 amino acids [17]. The crucial role of a single amino acid, the non-conserved tryptophan 38 (W38) for ALV-J binding, has been demonstrated previously in vitro [8,18,19]. Based on these findings, the chNHE1 modification in the generated chicken line consists of a single amino acid deletion of tryptophan 38 (W38) [14,15]. In both studies, ALV-J resistance was assessed utilizing a GFP-transducing [14,15] or a v-*src*-transducing [14] subgroup-J specific RCAS (replication-competent avian sarcoma-leukosis virus with a splice acceptor) reporter virus. While this is a valuable system to evaluate ALV-J susceptibility, it does not fully represent field conditions [18]. Therefore, a validation of this approach using the pathogenic ALV-J prototype virus to infect genetically modified commercial chickens is required to evaluate the efficacy of this strategy for the poultry industry.

In this study, we infected genetically modified commercial chickens harboring the NHE1 mutation (NHE1ΔW38). We could demonstrate that replication of the ALV-J field strain HPRS-103 is severely impaired in NHE1ΔW38 chickens and that no antigen was detectable in cloacal swabs at later time points. In addition, no signs of disease were observed in histopathology. Taken together, our study shows that the NHE1 mutation can prevent disease in commercial chickens and provides a practical approach to combat this deadly pathogen.

## 2. Materials and Methods

### 2.1. Ethics Statement

We conducted all animal work in compliance with relevant national and international guidelines for care and humane use of animals. The animal use protocol for the study reported here was approved by the Landesamt für Gesundheit und Soziales (LAGeSo) Berlin, Germany (approval number: G 0032/20).

### 2.2. Animals

The genetically modified chicken line was established recently [15]. The genetic background of these birds is a widely used, commercial egg-type chicken line (White Leghorn line; Lohmann selected Leghorn) obtained from Lohmann-Tierzucht GmbH (Cuxhaven, Germany). Chickens that express the NHE1W38 (NHE1W38 = W38^+/+^ and W38^+/−^ genotypes combined) and chickens that lack W38 (NHE1ΔW38, W38^−/−^) were hatched and housed in enriched cages in a S2 animal facility at the Freie Universität Berlin (Center for Infection Medicine). Animals were handled under strict biosecurity measures rooms with HEPA-filtered air. Infected edited and non-edited animals were co-housed in the same cages during the entire experiment to provide identical conditions. Water and commercial feed were provided ad libitum. Genotyping was conducted as previously described [15].

### 2.3. In Vivo Infection

To assess the genetically introduced ALV-J resistance of NHE1ΔW38 birds, we challenged these animals with the highly pathogenic HPRS-103 strain (nucleotide sequence accession number Z46390.1), kindly provided by the Pirbright Institute. The virus was propagated in chicken embryo fibroblasts, harvested from cell-free medium, and stored at −80 °C. One-day-old NHE1W38 chickens (W38^+/+^ *n* = 7; W38^+/−^ *n* = 11) and NHE1ΔW38 chickens (W38^−/−^ *n* = 11) were infected intraperitoneally with 200 µL HPRS-103 (10^5^ TCID50) as described previously [19,20]. All infected chickens were housed together. Non-infected birds (*n* = 3; NHE1W38^+/+^, NHE1W38^+/−^, and NHE1W38^−/−^) were kept separately. Peripheral whole blood samples (NHE1W38 *n* = 7; NHE1ΔW38 *n* = 8) were collected from infected chickens at 7, 10, 14, 21, 28, 35, 49, 63, and 77 days post infection (dpi) to quantify ALV-J proviral DNA copies in blood using qPCR [21,22]. Chickens were monitored for clinical symptoms of an ALV-J infection throughout 91 days. All infected birds were euthanized and examined for tumor lesions after termination of the experiment. Serum was prepared from blood to investigate the production of ALV-J-specific antibodies using ELISA. Organs were stored at neutral buffered formalin and submitted to the pathology department to examine the presence of ALV-J-induced tumor cells in various organs.

### 2.4. Quantification of Proviral DNA Copies

Proviral DNA was extracted from blood of the infected NHE1W38 and NHE1ΔW38 chickens using the NucleoSpin 96-well Blood Core Kit (Macherey-Nagel, Düren, Germany) following the manufacturer’s instructions. ALV-J proviral DNA copies were determined by quantitative PCR (qPCR) using primers and a probe for the ALV-J gp85 gene as published previously [23]. gp85 copy numbers were normalized to the chicken cellular inducible nitric oxide synthase (iNOS) gene as described previously [24,25].

### 2.5. Serology and Analysis of Viral Shedding

The presence of ALV-J specific antibodies was determined with a commercial ALV-J gp85-specific ELISA system (IDEXX GmbH, Ludwigsburg Germany). Diluted plasma (7, 14, 21, 28, 35 dpi) and serum samples (49, 76, 91 dpi) were analyzed according to the manufacturer’s instructions. To investigate viral shedding, a commercial ALV p27-specific antigen ELISA system was used (IDEXX GmbH). Cloacal swabs were collected on 49, 63, and 91 dpi and stored in standard Dulbecco’s Modified Eagle Medium (Biochrom AG, Berlin, Germany) at −20 °C. Then, 100 µL of undiluted sample was used for measurements according to the manufacturer’s instructions.

All samples were tested in duplicates. Calculations were performed as indicated by the manufacturer’s instructions. The presence or absence of specific antigen or antibodies was determined by relating the OD (650 nm) of a test sample to a standardized positive control:sample(S)positive control (P)=sample (mean)−negative control (mean)positive control (mean)−negative control (mean)

Serum and plasma samples with values of S/P > 0.60 were considered as positive. Samples from cloacal swabs with values of S/P > 0.20 were considered as positive.

### 2.6. Histopathology

Bone marrow, liver, kidneys, heart, and gonads were harvested from the infected chickens, stored at room temperature in 4% neutral buffered formalin, and subsequently assessed for pathological lesions. Histopathological examination was conducted by the pathological lab service of the Bavarian Animal Health Services (Poing, Germany). Staining with hematoxylin and eosin (H&E) was performed on paraffin-embedded, 4 µm thick sections. To eliminate subjectivity, all histopathological analyses were performed in a blinded manner.

### 2.7. Statistical Analyses

Statistical analysis was carried out using SPSS24 statistics (version 24.0.0.0) software (IBM, Armonk, NY, USA). Normally distributed data (Shapiro-Wilk test *p* > 0.05) were analyzed by Student’s *t*-tests. Mann-Whitney U tests were applied for non-normally distributed data. Graphs were constructed using GraphPad Prism (version 8.0.1 145) (GraphPad Software, San Diego, CA, USA).

## 3. Results

To investigate whether the NHE1ΔW38 mutation mediates resistance against an ALV-J field strain in vivo, NHE1W38 and NHE1ΔW38 chickens were infected with the pathogenic HPRS-103. All chickens were kept in the same room to assess whether NHE1ΔW38 are protected against ALV-J even in a virus-contaminated environment.

To assess virus replication in the infected animals, blood was collected at 7, 14, 21, 28, 35, 49, 63, and 77 dpi and ALV-J proviral genome copies determined by qPCR. One-week post infection, ALV-J genome copies were comparable between both groups (Figure 1A). The viral copies in NHE1ΔW38 chickens severely decreased from 7 to 28 dpi, reaching almost zero at 63 and 77 dpi, while constantly high genome copies were observed in NHE1W38 birds.

Next, we determined whether the virus is shed from infected animals. Cloacal swabs were analyzed at 49, 63, and 91 dpi using a commercial p27 ALV-specific antigen ELISA system. Constant shedding of ALV-specific p27 was observed in NHE1W38 birds, while no shedding was observed in NHE1ΔW38 birds (Figure 1B). Shedding of subgroup-J-specific antigen was confirmed in three swabs of NHE1W38 birds by gp85-specific PCR. Taken together, our data revealed that proviral load is severely impaired and that no virus is shed from commercial chickens harboring the NHE1ΔW38 mutation.

To investigate whether the humoral immune response contributes to the decline in the viral load, we measured ALV-J-specific serum antibodies using a commercial gp85-specific ELISA system. During the sharp decline of the ALV-J viral load in NHE1ΔW38 chickens until 21 dpi, no ALV-J-specific antibodies were detected (Figure 1C). At the end of the experiment, 54% of NHE1ΔW38 and 66% of NHE1W38 chickens had detectable antibody titers (S/P > 0.6). Relative antibody titers were higher in the NHE1W38 group.

Finally, all chickens were sacrificed at 91 dpi and screened for pathological macroscopic and microscopic lesions indicative of myeloid leukosis (ML). Animals were also weighed revealing that the absolute body weights of NHE1W38 birds were significantly lower compared to NHE1ΔW38 birds (Figure 2A). In addition, livers of NHE1W38 chickens were significantly larger compared to NHE1ΔW38 birds (Figure 2B). One of the NHE1W38 chickens displayed a solid liver tumor (Figure 2C), while no macroscopic tumors were detected in the NHE1ΔW38 birds. These macroscopic findings in NHE1W38 birds are in line with typical symptoms of an early stage of ALV-J infection.

Histopathological evaluations of ALV-J-induced lesions were performed on bone marrow, liver, kidney, gonads, and heart (Figure 3). Lymphocytic lesions indicative of an ALV-J infection were observed in all screened organs obtained from NHE1W38 chickens. Bone marrow and kidney were most affected organs with a frequency of lymphocytic infiltrations of 28% and 72%, respectively (Figure 3A). Lymphocytic lesions were not observed in NHE1ΔW38 chickens.

Taken together, clear signs of ALV-J infection and virus replication were observed in NHE1W38 birds while this was not the case in the genetically modified NHE1ΔW38 chicken line. Data obtained from serological and pathological examinations provide clear evidence that the NHE1W38 deletion mediates ALV-J resistance against the highly pathogenic ALV-J prototype strain HPRS-103, highlighting that this strategy for disease control in commercial chickens could be used in the field.

## 4. Discussion

Until now, no commercial vaccines are available to protect poultry against ALV-J. Recently generated chickens harboring the NHE1ΔW38 mutation are valuable models to investigate alternative control strategies against ALV-J infection [14,15]. However, evidence was still missing that this strategy also works in commercial chicken lines and upon infection with a pathogenic ALV-J strain. In this study, we provide evidence that NHE1ΔW38 mutation mediates resistance against HPRS-103, the prototypic ALV-J field strain, in a commercial chicken line.

Infected chickens were monitored for 13 weeks after infection according to standard laboratory tests to assess the viral load, virus shedding, and ALV-J-specific antibodies [26]. At final necropsy, our pathological evaluation focused on typical symptoms commonly exhibited at an early stage of ML. This was done to minimize distress of infected animals due to an increase in tumor mortality expected around 20 weeks post-infection [25].

The severe impairment of ALV-J replication in NHE1ΔW38 chickens directly correlated with the lack of viral shedding, as no antigen was detected in cloacal swabs. However, the relatively high proviral load of ALV-J in serum of NHE1ΔW38 chickens during the first weeks post infection is contradictory. Virus binding to an alternative receptor is unlikely, based on the previous findings that NHE1W38 is both necessary and sufficient for virus entry [14,15,18,19]. More conceivable is the binding of ALV-J to the modified receptor with reduced affinity. The uptake of virus particles by endocytosis could be another explanation, especially after infection when high levels of the inoculated virus are circulating. Future research will elucidate the nature of the proviral ALV-J copies in the blood of infected NHE1ΔW38 chickens and clarify whether this proviral DNA is extrachromosomal or chromosomally integrated. From what we observed in this study, however, the severe decline in the viral load over time indicates that no resistant ALV-J infection emerged. Even though all animals were infected, some NHE1ΔW38 and NHE1W38 chickens did not produce ALV-J-specific antibodies.

The final necropsy revealed a typical pathological picture of an ALV-J infection in susceptible NHE1W38-expressing birds including growth retardation, liver enlargement, and cell accumulations in multiple organs. Consistent with our findings, the liver, kidney, and bone marrow are the commonly and predominantly affected organs [27]. Lymphocytic infiltrations display the main histopathological finding in susceptible birds. This differs from the classical picture of ML characterized by myeloid cell infiltrations [25]. Even though ML represents the primary ALV-J-induced disease in meat-type chickens [28], a variety of other tumors such as hemangiomas, nephromas, and lymphosarcomas are accompanied with ALV-J depending on the virus strain or chicken breed [29].

The long-lasting ability to inhibit virus replication by the NHE1ΔW38 mutation remained a major concern. By cohousing susceptible and resistant chickens, we investigated the stability of resistance under selective pressure. In this setup, NHE1ΔW38 birds were continuously exposed to ALV-J virions, shed by susceptible NHE1W38 chickens. This may favor virus evolution and the emergence of viruses that may use alternative receptors or bind to the altered chNHE1 structure. However, no escape viruses were detected in NHE1ΔW38 chickens in our experiment, suggesting that the introduced conformational changes in the host cell receptor cannot be easily overcome. This is also supported by the fact that NHE1ΔW38 cells are resistant to distinct ALV-J subsets [14] and that the NHE1ΔW38 receptor variant is present in chukar partridge (*Alectoris chukar*), a widely distributed, ALV-J resistant wild bird [18].

Virus–host co-evolution continuously changes the conditions of virus–host interaction through sequences of random mutations [30]. Neither genome engineering, nor traditional control strategies can ensure a permanent antiviral barrier. However, inhibition of virus replication in a host population can significantly reduce the chance of viral adaption. Therefore, protective measures either preventing virus entry or inhibiting virus genome replication are particularly valuable. In this context, gene editing provides an important alternative to vaccination [31].

While targeted genome editing has been proven useful to study infectious diseases in chickens [32], our ALV-J resistant chickens can contribute to improved animal health worldwide and facilitate the alleviation of economic damage. The strategy is transferable to other chicken breeds as well as turkeys. Lastly, genetic engineering promotes the expansion of targeted analyses of virus–host interaction and can overcome limits of classical trait-selective breeding.

## Figures and Tables

**Figure 1 microorganisms-09-01066-f001:**
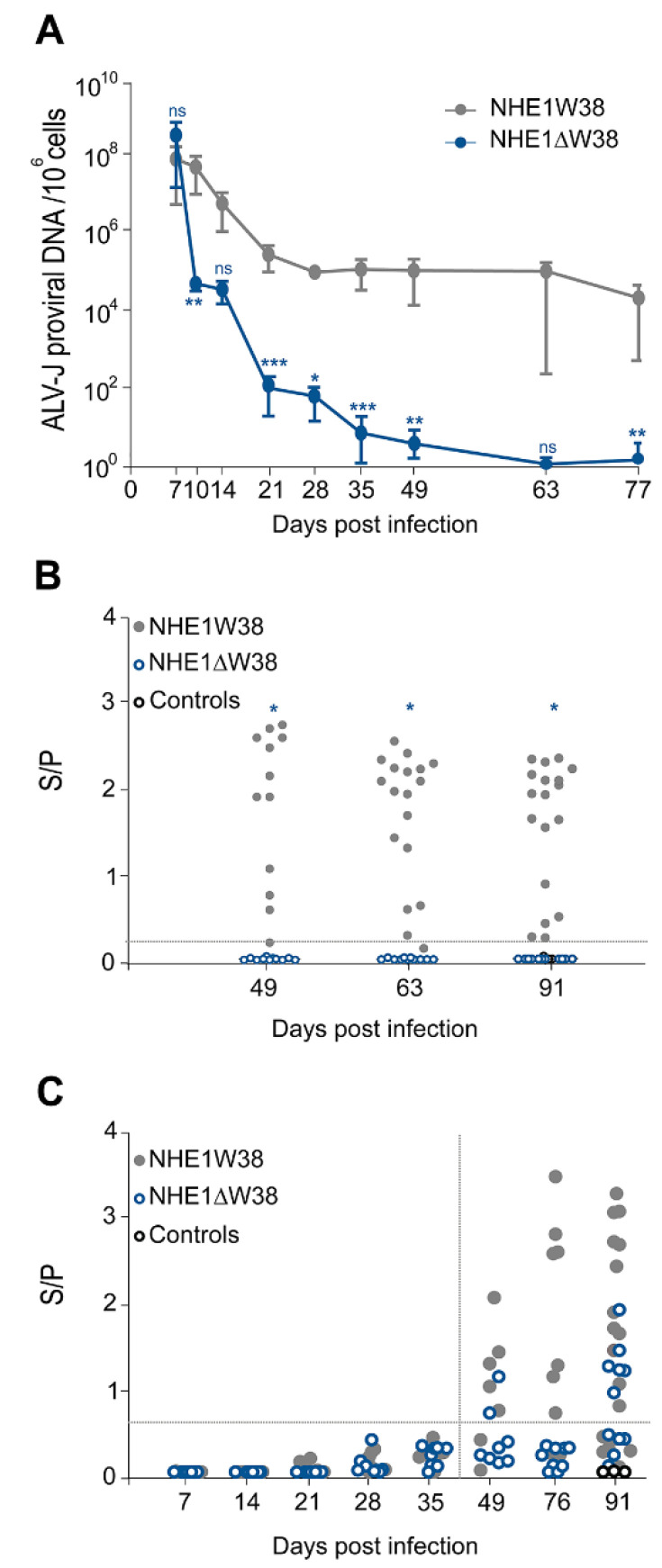
Serological analysis of ALV-J infection. (**A**) Quantitative analysis of ALV-J proviral genome copies by qPCR to determine viremia in infected NHE1ΔW38 (*n* = 8) and NHE1W38 birds (*n* = 7). Genome copies are displayed relative to 10^6^ cellular copies from 7 to 77 dpi. Error bars indicate standard deviation (SD). Significant differences are indicated by asterisk(s) (* *p* < 0.05, ** *p* < 0.01, *** *p* < 0.001; multiple Mann–Whitney U test). (**B**) Detection of ALV p27-specific antigen in cloacal swabs from 49, 63, and 91 dpi by ELISA (*n* ≥ 11). S/P represents sample (S) to positive control (P) ratio that indicates the relative level of antigen. S/P > 0.20 were considered as positive. Significant differences are indicated by asterisk (*p* < 0.05; Mann–Whitney U test). Non-infected birds served as control. (**C**) Detection of ALV-J gp85-specific plasma (7–35 dpi) and serum (49–91 dpi) antibodies by ELISA (*n* ≥ 7). S/P represents the relative antibody titer. S/P > 0.60 were considered as positive. Statistical analysis of 7–49 dpi was performed using the Mann–Whitney U test (*p* > 0.05) and a two-sided Student’s t-test was used for statistical analysis of 76 and 91 dpi (*p* > 0.05). Non-infected birds served as control.

**Figure 2 microorganisms-09-01066-f002:**
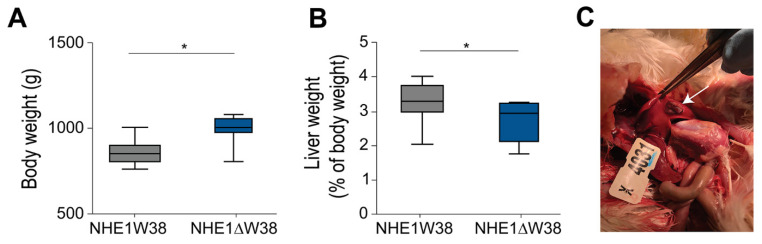
ALV-J infection leads to typical pathological lesions in NHE1W38 animals. (**A**) Body weights of infected NHE1W38 and NHE1ΔW38 birds (*n* ≥ 9) at 91 dpi. Statistical analysis was performed using a two-sided Student’s t-test compared to NHE1W38 birds (* *p* < 0.05). The median is shown as a central line of the box. Boxes and whiskers represent data distribution. (**B**) Relative liver weights of infected NHE1W38 chickens and NHE1ΔW38 birds on 91 dpi (*n* ≥ 9). Statistical analysis was performed using a two-sided Student’s t-test compared to NHE1W38 birds (* *p* < 0.05). (**C**) ALV-J typical gross liver tumor in a NHE1W38 bird at 91 dpi as indicated by arrow.

**Figure 3 microorganisms-09-01066-f003:**
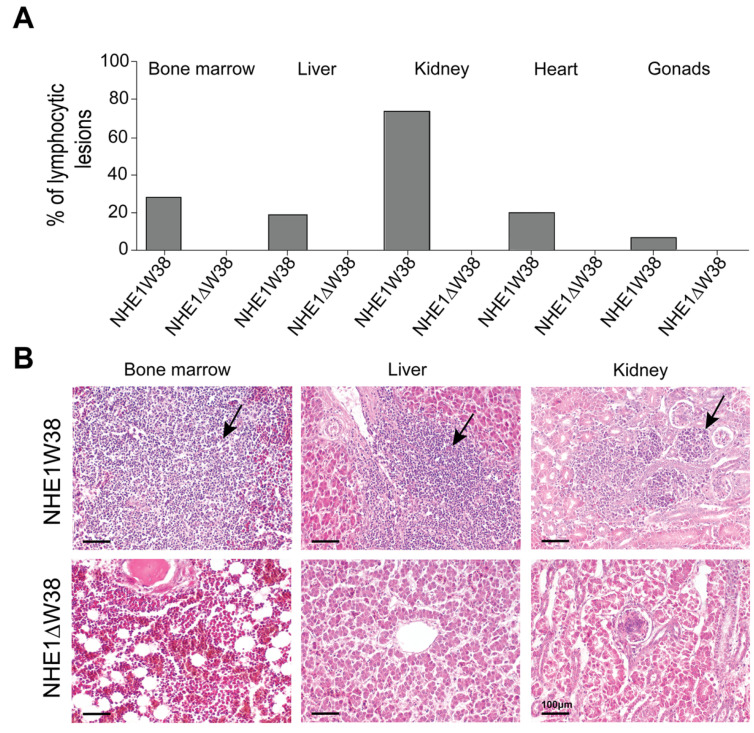
Assessment of ALV-J-related histopathological lesions. (**A**) Frequencies of lymphocytic lesions observed in bone marrow, liver, kidney, heart, and gonads of NHE1W38 chickens and NHE1ΔW38 birds. The highest frequency was found in the kidney with 72%. (**B**) Light microscopy of representative hematoxylin and eosin (H&E)-stained sections from bone marrow, liver, and kidney of NHE1W38 and NHE1ΔW38 birds. Arrows show lymphocyte infiltrations in all three organs of NHE1W38 chickens. No alteration or infiltration is present in the organs of NHE1ΔW38 birds. Scale bar = 100 µm.

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
