# Peer review of "A Genetically Engineered Commercial Chicken Line Is Resistant to Highly Pathogenic Avian Leukosis Virus Subgroup J"

_microorganisms, 2021, doi:10.3390/microorganisms9051066_

Round 1
Reviewer 1 Report
The manuscript entitled “A Genetically Engineered Commercial Chicken Line 2 is Resistant to Highly Pathogenic Avian Leukosis Virus 3 Subgroup J” by Kheimar et al. describes the findings of in vivo experimental infection of genetically modified commercial chickens with pathogenic ALV-J prototype virus. Using CRISPR/Cas9 as genome editing technology to modify host genes that play a key role in the viral replication cycles represents an interesting topic especially in case of lack of efficient traditional control strategies as in case of ALV-J virus.
The manuscript is well-written and represents sufficient preliminary data about the impact of mutation of the chicken ALV-J receptor Na+/H+ exchanger type 1 (chNHE1) on the ALV-J pathogenicity. Therefore, I suggest a minor revision.
Minor points:
Introduction: more data about ALV-J virus entry and the importance of chNHE1 is required in the introduction section.
Line 85: did you test the edited and non-edited one-day-old chicks for other possible concurrent infection and status of the maternal immunity. If yes please add a brief.
Line 93: please add the accession number of the HPRS-103 strain.
Line 92-94: please add more detail about the Animal Experiments, as housing condition, groups allocation, negative control groups housing, and number, specify whether you have both edited and non-edited chickens negative control groups?
Line 96: please add the number of replicates used to collect the Peripheral blood samples.
Line 97: please specify whether the DNA has been extracted from the whole blood or buffy coat. And add a reference for the procedure.
Line 98: please specify the clinical sign index that has been used with a reference.
Line 110: please add a reference for (iNOS) gene.
Line 116: please add the sampling time of the cloacal swabbing.
Line 125: please specify which organs that have been collected.
Figure 2.A. please add the bodyweight of the non-infected birds.
Author Response
We thank the reviewer for the valuable comments/suggestions. Please find our point by point response enclosed.
- Introduction: more data about ALV-J virus entry and the importance of chNHE1 is required in the introduction section.
As suggested by the reviewer, we now provide additional data on ALV-J entry and the importance of chNHE1 in the introduction section (L63-67).
- Line 85: did you test the edited and non-edited one-day-old chicks for other possible concurrent infection and status of the maternal immunity. If yes please add a brief.
Thanks for this interesting comment. The breeding and animal experiment were performed in animal facilities with high biosafety standards, which are frequently monitored for specific pathogens based on the microbiological status of the animals. Therefore, a separate analysis of concurrent infections and the immunological status was not performed in our experiment (L95-97).
- Line 93: please add the accession number of the HPRS-103 strain.
As suggested by the reviewer, we now provide the accession number of HPRS-108 strain in (L101,102).
- Line 92-94: please add more detail about the Animal Experiments, as housing condition, groups allocation, negative control groups housing, and number, specify whether you have both edited and non-edited chickens negative control groups?
Thanks for raising this point. We included the requested information in the manuscript (L95-97).
- Line 96: please add the number of replicates used to collect the Peripheral blood samples.
Thanks for this observation. The number of replicates is now provided in the text (L 105) and the figure legend (LL158-159).
- Line 97: please specify whether the DNA has been extracted from the whole blood or buffy coat. And add a reference for the procedure.
Thanks for picking this up. We clarified this point and added the reference (LL105,107).
- Line 98: please specify the clinical sign index that has been used with a reference.
The infected animals did not develop any clinical symptoms during the course of infection. Lesions were only observed after final necropsy.
- Line 110: please add a reference for (iNOS) gene.
As suggested, we included the reference for the iNOS gene (L122).
- Line 116: please add the sampling time of the cloacal swabbing.
Thanks for this comment. We added the sampling time of the cloacal swabs in the text (L128) and the figure legend (L165).
- Line 125: please specify which organs that have been collected.
As suggested by the reviewer, we now specify the collected organs in the manuscript (L139).
- Figure 2.A. please add the bodyweight of the non-infected birds.
Thanks for pointing this out. In our experiment, we only had very few non-infected animals but the body weighs of non-infected birds have been extensively characterized in a previous report (DOI: 10.3389/fgeed.2020.00003). The mean body weight of the infected NHE1DW38 animals was comparable to the uninfected animals in this study. This study focuses on a direct comparison between infected edited and non-edited birds.
Reviewer 2 Report
In their manuscript “A Genetically Engineered Commercial Chicken Line...“ Kheimar et al. present a short follow-up study on the resistance of genetically engineered chickens to ALV-J, a considerable pathogen of domestic poultry. The virus resistance of chickens bearing single amino-acid deletion in NHE1, the specific ALV-J receptor, has been described experimentally in two independent studies and the present contribution could be regarded as incremental. On the other hand, it shows that the previous conclusions obtained with recombinant virus vectors are valid for the infectious HPRS103 ALV-J strain, whose application simulates the real field conditions. Although the results are fully convincing, I have some concerns to be specified before final consideration.
- Authors applied the virus intraperitoneally. This probably is not the natural way how the virus infects the host. I would like to have some discussion on how it affects the virus persistence, pathogenesis, and the host immune response.
- I am surprised by the high provirus load 7 dpi and relatively long-term presence of proviral copies in the blood of infected NHE1DW38 chickens (Fig. 1A). This is in contrast to the previous in vitro infections with cell-free virus. In resistant animals with inactive receptor, one would expect at least decreased load of HPRS103 DNA. The nature of the virus DNA (integrated or extrachromosomal) remains enigmatic and authors should examine not only the presence of DNA, but also the infectious virus in circulation (virus propagation, qRT-PCR and/or PERT assay).
- I miss the late effects of HPRS103. All animals were sacrificed 13 weeks after infection, which makes impossible to say if the histopathological lesions are completely absent or just delayed in NHE1DW38 animals. Furthermore, provided that the organ samples were stored, detection of virus DNA in histopathological lesions would be illustrative.
- In methods, I miss the description of HPRS103 propagation and a better description of the white leghorn chickens used for gene editing and virus infection experiments. Was it meat- or egg-type? It is important because HPRS103, in contrast to later Chinese strains, used to exert pathogenic activity only in broilers.
- Authors should explain the use of iNOS gene to normalize qPCR (L109). It would be better to normalize qPCR to any HS gene with constitutive expression rather than to an inducible marker whose expression would certainly fluctuate with lymphocyte infiltration and possibly with virus load.
- In Figure 3A, it is problematic to describe the results as lymphocytic infiltration in bone marrow, where we would expect the presence of lymphocyte precursors in both NHE1W38 and NHE1DW38 chicken groups. However, if authors evaluate the results against the background of the bone marrow, then it would be more appropriate to use the term e.g. % of lymphocytic lesions to describe the y-axis, which more accurately describes the histopathological findings. The results of histopathological evaluation would be supported by immunohistochemical analysis (IHC) and/or in situ hybridization for viral nucleic acid detection.
- In results, authors should also explain the dynamics of the production of specific antibodies against gp85 in NHE1DW38 chickens in the interval from 49 to 91 DPI (Fig. 1C). The lower titer of specific antibodies in NHE1DW38 chickens is explained by the shorter duration of virus exposure as a result of impaired virus replication, but this claim is not experimentally confirmed by detection of virus in circulation (virus propagation, qRT-PCR).
- In discussion (L228-231), the authors also hypothesize that immune tolerance develops due to a possible early horizontal infection in some NHE1DW38 and NHE1W38 chickens where, however, no specific antibodies were detected. How do the authors explain the development of immune tolerance in resistant individuals NHE1DW38 with an inactive receptor?
9. In methods, section 2.5. Serology and Analysis of viral shedding (line 123-124), it is stated that samples of serum and plasma with S/P values ​​greater than 0.2 were considered positive and samples from cloacal swabs with S/P values ​​greater than 0.6 as well. In contrast, in Fig. 1B and Fig. 1C, the thresholds are just the opposite. Furthermore, the author should justify why they used plasma obtained from anticoagulated blood to detect specific antibodies up to 35 DPI and from 49 to 91 DPI used serum from coagulated blood. Maybe it would be more appropriate to compare results from plasma and serum in two plots.
typos:
L148: 106 – exponent missing
Author Response
We thank the reviewer for the valuable comments and suggestions. Please find our point by point response enclosed.
1. Authors applied the virus intraperitoneally. This probably is not the natural way how the virus infects the host. I would like to have some discussion on how it affects the virus persistence, pathogenesis, and the host immune response.
As suggested by the reviewer, we now discuss these points. There are multiple ways of ALVJ experimental infection which are well established in ALV-J research. We added the references for i.p. methods in L106.
2. I am surprised by the high provirus load 7 dpi and relatively long-term presence of proviral copies in the blood of infected NHE1DW38 chickens (Fig. 1A). This is in contrast to the previous in vitro infections with cell-free virus. In resistant animals with inactive receptor, one would expect at least decreased load of HPRS103 DNA. The nature of the virus DNA (integrated or extrachromosomal) remains enigmatic and authors should examine not only the presence of DNA, but also the infectious virus in circulation (virus propagation, qRT-PCR and/or PERT assay).
Thanks for your comment. We were also puzzled by the high proviral load at 7 dpi, especially as proviral copies require entry of the virus. This should theoretically not be possible in the absence of the receptor, unless there is an alternative receptor. We would like to address this phenomenon in a future study as additional ex vivo samples/animal experiments are required.
3. I miss the late effects of HPRS103. All animals were sacrificed 13 weeks after infection, which makes impossible to say if the histopathological lesions are completely absent or just delayed in NHE1DW38 animals. Furthermore, provided that the organ samples were stored, detection of virus DNA in histopathological lesions would be illustrative.
Thanks for the great suggestion. We tried to recover DNA for qPCR from the formalin fixed tissues, but did not succeed due to the long-term formalin fixation.
4. In methods, I miss the description of HPRS103 propagation and a better description of the white leghorn chickens used for gene editing and virus infection experiments. Was it meat- or egg-type? It is important because HPRS103, in contrast to later Chinese strains, used to exert pathogenic activity only in broilers.
As suggested by the reviewer, we now provide the description of HPRS103 propagation as well as the white leghorn chickens (L89-91 and L102-103).
5. Authors should explain the use of iNOS gene to normalize qPCR (L109). It would be better to normalize qPCR to any HS gene with constitutive expression rather than to an inducible marker whose expression would certainly fluctuate with lymphocyte infiltration and possibly with virus load.
Thanks for raising this point. We apparently failed to convey that the iNOS gene (not RNA) was only used to normalize proviral DNA copies to the number of host cells. For qRT-PCR, HS genes with constitutive expression are certainly the right choice as pointed out by the reviewer.
6. In Figure 3A, it is problematic to describe the results as lymphocytic infiltration in bone marrow, where we would expect the presence of lymphocyte precursors in both NHE1W38 and NHE1DW38 chicken groups. However, if authors evaluate the results against the background of the bone marrow, then it would be more appropriate to use the term e.g. % of lymphocytic lesions to describe the y-axis, which more accurately describes the histopathological findings. The results of histopathological evaluation would be supported by immunohistochemical analysis (IHC) and/or in situ hybridization for viral nucleic acid detection.
As suggested by the reviewer, we changed the y-axis label to “% of lymphocytic lesions” and emphasized this also in the figure legend (L208-209).
7. In results, authors should also explain the dynamics of the production of specific antibodies against gp85 in NHE1DW38 chickens in the interval from 49 to 91 DPI (Fig. 1C). The lower titer of specific antibodies in NHE1DW38 chickens is explained by the shorter duration of virus exposure as a result of impaired virus replication, but this claim is not experimentally confirmed by detection of virus in circulation (virus propagation, qRT-PCR).
Thanks for pointing this out. We agree with the reviewer and have removed this claim as it is not experimentally confirmed (L239).
8. In discussion (L228-231), the authors also hypothesize that immune tolerance develops due to a possible early horizontal infection in some NHE1DW38 and NHE1W38 chickens where, however, no specific antibodies were detected. How do the authors explain the development of immune tolerance in resistant individuals NHE1DW38 with an inactive receptor?
We agree with the reviewer and have therefore removed the hypothesis of immune tolerance induction (L241-243).
9. In methods, section 2.5. Serology and Analysis of viral shedding (line 123-124), it is stated that samples of serum and plasma with S/P values ​​greater than 0.2 were considered positive and samples from cloacal swabs with S/P values ​​greater than 0.6 as well. In contrast, in Fig. 1B and Fig. 1C, the thresholds are just the opposite.
Thanks for catching this error. We corrected it as indicated (LL 136-137).
Furthermore, the author should justify why they used plasma obtained from anticoagulated blood to detect specific antibodies up to 35 DPI and from 49 to 91 DPI used serum from coagulated blood. Maybe it would be more appropriate to compare results from plasma and serum in two plots.
Thanks for raising this question. During the early time points we only collected small blood volumes (40 uL) due to the size of the chickens that were not suitable for serum collection. As the birds grew larger, we collected sufficient amounts of blood for serum preparation. We now separated the two datasets “plasma vs. serum” by a dashed line to address the reviewer’s comment see figure 1C.
typos:
L148: 106 – exponent missing
Thanks, we corrected it L162.
Round 2
Reviewer 2 Report
In contrast to sound answers to my concerns, the high proviral load of ALV-J detected in blood samples of chickens with the NHE1∆W38 receptor has not been satisfactorily explained. The authors hypothesize the presence of an alternative receptor, which does not fit with the previous findings that NHE1 is both necessary and sufficient for virus entry. How do the authors predict the role of this alternative receptor when the proviral copy number of NHE1∆W38 chickens has gradually decreased throughout the experiments? Is the proviral DNA extrachromosomal only? Has any integration site been detected? I would expect more detailed discussion on this striking contradiction. Without a proof of infectious virus, any statement that NHE1∆W38 chickens in mixed flock were exposed and resisted transmission from wt animals is vague, especially when non-infected negative control animals were present.
Author Response
We thank the reviewer for pointing this out.
As suggested, we included a more detailed discussion about the high proviral load in blood samples of chickens with the NHE1∆W38 receptor (L238-244). We do not preferably assume that this can be explained by binding to an alternative receptor.
Thanks for your comment concerning the non-infected negative control animals. These birds were kept separately. We added this information in line 106-107.
Based on the continuous detection of ALV-J antigen in cloacal swabs of infected WT birds by the ALV-specific p27-ELISA, we assume that NHE1∆W38 chickens were exposed to ALV-J.